# Decision-Making Evaluation of the Pilot Project of Comprehensive Land Consolidation from the Perspective of Farmers and Social Investors: A Case Study of the Project Applied in Xianning City, Hubei Province, in 2020

**Wei Xia and Gangqiao Yang \***

School of Public Administration, Huazhong Agricultural University, Wuhan 430070, China
\* Correspondence: ygqygq@webmail.hzau.edu.cn

**Abstract:** Comprehensive land consolidation is an important means to implement the rural revitalization strategy. The decision-making of comprehensive land consolidation projects is the basis of scientifically selecting land consolidation projects, ensuring the quality of project, and making the project advance in an orderly manner. Compared with the traditional land consolidation project, the overall land consolidation project has a large demand for funds, and the participation of social capital has become an important way to solve the project funding problem. From the perspective of farmers and social investors, this research constructs a comprehensive land consolidation project decision-making evaluation index system and evaluation method from five aspects, including agricultural land consolidation, construction land consolidation, rural ecological protection and restoration, rural historical and cultural protection, and rural industrial development goals. The results show that there is a big difference in the evaluation results from the perspective of farmers and social investors. Considering the urgency of farmers' needs and the investment willingness of social investors in comprehensive land consolidation, the evaluation results are basically consistent with the actual project approval. The index system and evaluation method established in this study are helpful to scientifically select pilot projects of comprehensive land consolidation and invest limited government financial funds into the consolidation contents that are both urgently needed by farmers and willing to be invested by social investors.

**Keywords:** comprehensive land consolidation; pilot project; decision-making evaluation; farmers; social investors

## 1. Introduction

With the tightening of resource and environmental constraints, problems such as disordered spatial distribution of rural land, inefficient use of resources, and deterioration of the ecological environment have become increasingly prominent [1–3]. The traditional land consolidation model that takes a single element as the consolidation object has been unable to cope with the continuous comprehensive problems in the process of rural development [4,5]. Under the background of the rural revitalization strategy, land consolidation has expanded from the single agricultural land consolidation to the comprehensive consolidation of the whole elements of "mountains, rivers, forests, fields, lakes, grasses and sand" [6–10]. The overall promotion of comprehensive land consolidation will help gradually narrow the gap between urban and rural development, stimulate the internal driving force of rural development, coordinate the harmonious development of man and nature, and ultimately achieve comprehensive rural revitalization [11–14].

As an important part of land consolidation project management, the decision-making is the basis of scientifically selecting land consolidation projects, ensuring the quality of project, and making the project advance in an orderly manner [15,16]. In recent years,

China's annual investment in land consolidation has reached hundreds of billions of Yuan [17]. However, due to the characteristics of large capital demand and long return time for comprehensive land consolidation, the supply of consolidation funds is still difficult to meet its demand. Western countries also face the limitation of funds [18], so they will strictly allocate the limited funds to the most suitable areas in the land consolidation project initiation stage to ensure effective resource management and successful financial support [2,19]. In order to solve the problem of the shortage of funds for comprehensive land consolidation, the Ministry of Natural Resources strongly advocates and encourages social capital to participate in comprehensive land consolidation and ecological restoration, and local governments also actively explore ways to attract social capital to comprehensive land consolidation. In the case of insufficient government financial funds, how to leverage or attract social capital to participate and ensure the high-quality implementation of comprehensive land consolidation projects has become an important issue to be solved in the decision-making of comprehensive land consolidation projects.

To attract social capital to participate in comprehensive land consolidation projects, the most important thing is to understand the interests of social investors and set up projects with high investment willingness of social investors so as to attract investment from social investors. At present, the academic research on the decision-making of land consolidation project mainly starts from the perspective of land [9,19,20], and there is relatively little literature on the decision-making of land consolidation projects from the perspective of the microsubject of the social capital. With the gradual development of land consolidation work, some scholars have found that mandatory land consolidation has adverse effects on farmers [21], and land consolidation should fully respect the dominant position of farmers [19,22–24]. In some places, in the process of social investors' participation in comprehensive land consolidation, the phenomenon of damage to the rights of farmers also appeared. Therefore, the decision-making of a comprehensive land consolidation project should not only consider the interests of the investor, namely the social capital, but also the rights of local farmers. Farmers are the ultimate beneficiaries of comprehensive land consolidation [25], and social investors are an important force to promote comprehensive land consolidation projects. It has important theoretical and practical significance to construct a comprehensive land consolidation project decision-making evaluation index system from the perspective of farmers and social investors.

Based on the perspective of farmers and social investors, this paper constructs a comprehensive land consolidation project decision-making evaluation index system and evaluation method. We performed an empirical analysis by using the survey data of seven pilot projects of comprehensive land consolidation in Xianning, Hubei Province, in 2020 and the entropy weight TOPSIS method. It provides the theoretical basis and case support for standardizing the decision-making of the pilot project of comprehensive land consolidation and promoting the pilot work of comprehensive land consolidation.

## 2. Materials and Methods

### 2.1. Study Area and Data Sources

In April 2020, the Office of the Leading Group for Comprehensive Land Consolidation of Hubei Province issued the "Notice on Application for Comprehensive Land Consolidation Projects", requiring the province to carry out the application of comprehensive land consolidation projects. Xianning City, located in the Wuhan urban circle, organized the following 7 projects to apply for the 2020 Hubei Province Comprehensive Land Consolidation Pilot Project: Zhaoliqiao Town Project in Chibi City (Project A), Henggouqiao Town Project in Xian'an District (Project B), Xiangyanghu Town Project in Xian'an District (Project C), Dupu Town Project in Jiayu County (Project D), Daping Township Project in Tongcheng County (Project E), Tiancheng Town Project in Chongyang County (Project F), Honggang Town Project in Tongshan County (Project G) (see Appendix A: Figure A1 and Table A1). This paper takes 7 applied projects as examples to conduct empirical research (Figure 1).

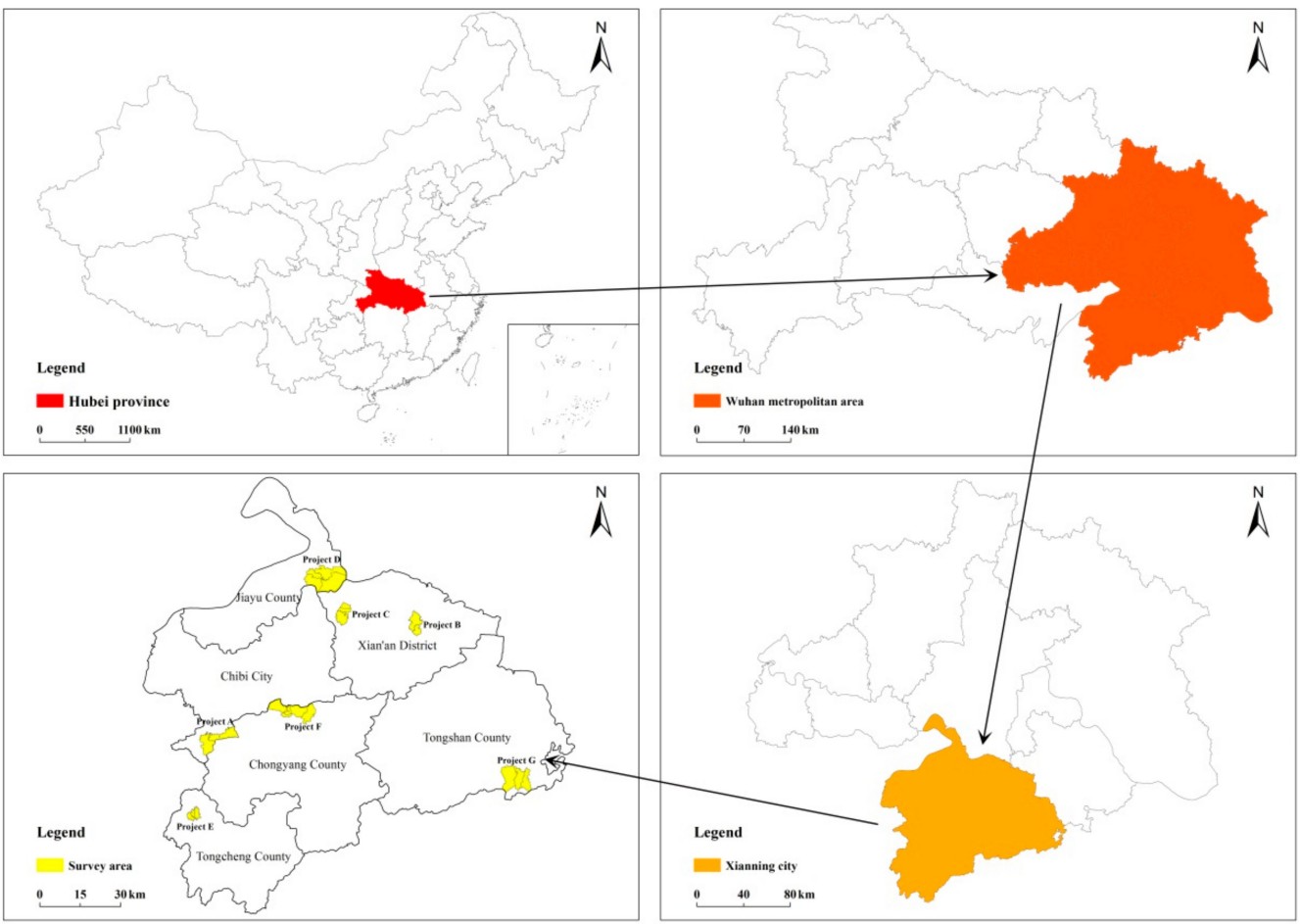

**Figure 1.** Survey area.

In order to obtain empirical data, the research group of more than 10 people conducted a questionnaire survey on farmers and social investors (new business entities) in the above-mentioned 7 project areas in January 2022. After removing invalid questionnaires, a total of 306 valid farmers questionnaires were obtained, including 47 items A, 35 items B, 45 items C, 47 items D, 43 items E, 41 items F, and 48 items G; and 20 valid social investor questionnaires, including 3 items A, 2 items B, 2 items C, 4 items D, 3 items E, 3 items F, and 3 items G (The details see Appendix A: Tables A2 and A3).

### 2.2. Research Methods

The entropy method is suitable for determining the weight of each index in the multi-index comprehensive evaluation. Because it calculates the weight based on the information entropy, the result is more objective [26–28]. The TOPSIS method (the distance method between superior and inferior solutions) is suitable for decision analysis for multiple targets [29]. Firstly, standardize the data to obtain a normalized vector $r_{lz}$, and establish a normalized decision matrix R. The calculation formula is:

$$r_{lz} = \frac{x_{lz} - x_{min}}{x_{max} - x_{min}} \tag{1}$$

In the formula: $x_{lz}$ is the actual value of the $z$ index of project area $l$; $x_{max}$ and $x_{min}$ are the maximum and minimum value of the single index, respectively, where $l = 1, 2, \cdots, m$, $z = 1, 2, \cdots, n$.

Then, use the entropy method to calculate the index weight, and its calculation formula is:

$$E_z = -k \sum_{l=1}^{m} f_{lz} \ln f_{lz} \tag{2}$$

$$w_z = \frac{1 - N_z}{n - \sum_{z=1}^{n} N_z} \tag{3}$$

In the formula: $E_z$ represents the entropy value of the $z$ index, and $w_z$ represents the entropy weight coefficient of the $z$ index; information entropy $k = \frac{1}{\ln m}$; the characteristic proportion of the index $f_{lz} = \frac{r_{lz}}{\sum_{i=1}^{m} r_{lz}}$, assuming that when $f_{lz} = 0$, $f_{lz} \ln f_{lz} = 0$.

On the basis of the normalized decision matrix, the entropy weight coefficient was added to establish a weighted normalized decision matrix. The calculation formula is:

$$v_{lz} = w_z \cdot r_{lz} \tag{4}$$

Determining the positive ideal solution $V^+$ and the negative ideal solution $V^-$ according to $v_{lz}$, and calculating the distance $D_l^+$ from the evaluation vector to the positive ideal solution $V^+$ and $D_l^-$ from the evaluation vector to the negative ideal solution $V^-$, the calculation formula is as follows:

$$V^+ = \{max\ v_{lz} \mid z = 1, 2, \cdots, n\} = \{v_1^+, \quad v_2^+, \quad \cdots, \quad v_n^+\} \tag{5}$$

$$V^- = \{min\ v_{lz} \mid z = 1, 2, \cdots, n\} = \{v_1^-, \quad v_2^-, \quad \cdots, \quad v_n^-\} \tag{6}$$

$$D_l^+ = \sqrt{\sum_{z=1}^{n} \left(v_{lz} - v_z^+\right)^2}(l = 1, 2, \cdots, m) \tag{7}$$

$$D_l^- = \sqrt{\sum_{z=1}^{n} \left(v_{lz} - v_z^-\right)^2}(l = 1, 2, \cdots, m) \tag{8}$$

Finally, the closeness was calculated, and the formula is as follows:

$$C_l = \frac{D_l^-}{D_l^+ + D_l^-}; (l = 1, 2, \cdots, m) \tag{9}$$

In the formula: $0 \leqslant C_l \leqslant 1$, the smaller the closeness $C_l$, the lower the degree; the greater the closeness $C_l$, the higher the degree.

### 2.3. Construction of Evaluation Index System

The comprehensive land consolidation mainly includes agricultural land consolidation, construction land consolidation, rural ecological protection and restoration, rural historical and cultural protection, etc., and the consolidation contents of these four aspects all serve the rural revitalization, especially the rural industrial development. Therefore, starting from the above-mentioned four aspects of the consolidation contents and industrial development goals, this paper analyzes the interests of farmers and social investors and then constructs a comprehensive land consolidation project decision-making evaluation index system.

2.3.1. Evaluation Index System from the Perspective of Farmers

Through the investigation, it was found that the majority of farmers are eager to change the backward production and living conditions in rural areas through comprehensive land consolidation. The worse the production and living conditions are, the higher the farmers' expectations of the comprehensive land consolidation project will be. Therefore, this paper constructs the comprehensive land consolidation projects decision-making evaluation index system from the perspective of farmers from the following five aspects.

The urgency for agricultural land consolidation. In the process of agricultural land consolidation, the interests of farmers mainly include: improvement of the comprehensive

quality of existing paddy fields, transformation of dry land into paddy fields, improvement of the comprehensive quality of other agricultural land. This paper subdivides the urgency for agricultural land consolidation into the following three indicators: the urgency to improve the comprehensive quality of existing paddy fields, the urgency for transforming dry land into paddy fields, and the urgency to improve the comprehensive quality of other agricultural land. Among them, the urgency to improve the comprehensive quality of existing paddy fields includes four indicators: the completeness of paddy field irrigation facilities, the completeness of paddy field drainage and waterlogging facilities, the completeness of field road facilities, and the degree of paddy field fragmentation. The first three indicators are negative indicators, and the last one is a positive index. The urgency to transform dry land into paddy fields is represented by the difficulty of transforming dry land into paddy fields, which is a negative index. The urgency to improve the comprehensive quality of other agricultural land includes the urgency to improve the comprehensive quality of garden land, the urgency to improve the comprehensive quality of economic forest land, and the urgency to improve the comprehensive quality of the pond. The urgency to improve the comprehensive quality of the garden land is measured by the completeness of the irrigation facilities of the garden land and the degree of transportation convenience of the garden land. The urgency to improve the comprehensive quality of the economic forest land is measured by the completeness of the irrigation facilities of the economic forest land and the degree of transportation convenience of the economic forest land. The urgency to improve the comprehensive quality of the ponds is measured by the degree of siltation, the degree of leakage, the degree of irrigation convenience, and the degree of transportation convenience, all of which are negative indicators.

The urgency for construction land consolidation. In the process of construction land consolidation, the interests of farmers mainly include: improvement of rural infrastructure and public service facilities and efficient use of rural construction land. This paper subdivides the urgency for construction land consolidation into the following two indicators: the urgency to improve rural infrastructure and public service facilities and the urgency to achieve efficient use of rural construction land. Among them, the urgency to improve rural infrastructure and public service facilities is characterized by the completeness of rural infrastructure and the completeness of rural public service facilities. The urgency to achieve efficient use of rural construction land is characterized by the intensive utilization of rural construction land, all of which are negative indicators.

The urgency for rural ecological protection and restoration. In the process of rural ecological protection and restoration, the interests of farmers mainly include: ecological environment restoration and human settlements improvement. This paper subdivides the urgency for rural ecological protection and restoration into two indicators: the urgency to achieve ecological environment restoration and the urgency for human settlements improvement. Among them, the urgency for ecological environment restoration is represented by the degree of water pollution, soil pollution, soil erosion, mine environment damage, and vegetation degradation, all of which are positive indicators. The urgency for human settlements improvement is represented by the satisfaction degree of sanitary toilet renovation, domestic waste treatment, domestic sewage treatment, and village appearance, all of which are negative indicators.

The urgency for rural historical and cultural protection. Through the investigation, it was found that the majority of farmers are eager to protect and restore local rural historical and cultural resources through comprehensive land consolidation in order to develop rural leisure tourism. This paper subdivides the urgency for rural historical and cultural protection into the following four indicators: the richness of historical and cultural resources, the popularity of historical and cultural resources, the degree of destruction of historical and cultural relics, and the willingness to build village historiographers, all of which are positive indicators.

The urgency for industrial development. Through the investigation, it was found that the majority of farmers are eager to promote the development of local industries through

comprehensive land consolidation in order to achieve the goals of rural beauty, industrial prosperity, and prosperity. This paper subdivides the urgency for industrial development into the following two indicators: the willingness to develop large-scale agriculture and the willingness to develop rural secondary and tertiary industries. These two indicators are both positive indicators, that is, the stronger the farmers' willingness to develop large-scale agriculture and rural secondary and tertiary industries, the stronger the farmers' desire to promote the development of local industries through comprehensive land consolidation, and the higher the urgency for industrial development. The opposite is also true.

To evaluate the decision-making of the comprehensive land consolidation project from the perspective of farmers is to judge the priority of the project by measuring the urgency of farmers' needs for comprehensive land consolidation. The higher the urgency of farmers' needs, the higher the order of project approval. In order to accurately measure the urgency of farmers' needs, this paper adopts the Likert 5-level scale as a tool to measure the urgency. The specific calculation method is shown in Table 1.

**Table 1.** Decision-making evaluation index system of pilot projects of comprehensive land consolidation from farmers' perspective.

| Target Layer | Criterion Layer | Indicator Layer | Definition | Value |
|---|---|---|---|---|
| The urgency of farmers' needs for comprehensive land consolidation | The urgency for agricultural land consolidation (0.179) | The urgency to improve the comprehensive quality of existing paddy fields | The completeness of paddy field irrigation facilities X1 (0.066) | High degree of completeness—low degree of completeness | 1–5 |
| | | | The completeness of paddy field drainage and waterlogging facilities X2 (0.117) | High degree of completeness—low degree of completeness | 1–5 |
| | | | The completeness of field road facilities X3 (0.085) | High degree of completeness—low degree of completeness | 1–5 |
| | | | The degree of paddy field fragmentation X4 (0.032) | Low degree—high degree | 1–5 |
| | | The urgency to transform paddy fields from dry land | The difficulty of transforming dry land into paddy fields X5 (0.179) | High difficulty—low difficulty | 1–5 |
| | | The urgency to improve the comprehensive quality of other agricultural land | The completeness of the irrigation facilities of the garden land X6 (0.075) | High degree of completeness—low degree of completeness | 1–5 |
| | | | The degree of transportation convenience of the garden land X7 (0.054) | High degree—low degree | 1–5 |
| | | | The completeness of the irrigation facilities of the economic forest land X8 (0.033) | High degree of completeness—low degree of completeness | 1–5 |
| | | | The degree of transportation convenience of the economic forest land X9 (0.095) | High degree—low degree | 1–5 |
| | | | The degree of siltation of the pond X10 (0.053) | Low degree—high degree | 1–5 |
| | | | The degree of leakage of the pond X11 (0.066) | Low degree—high degree | 1–5 |
| | | | The degree of irrigation convenience of the pond X12 (0.078) | High degree—low degree | 1–5 |
| | | | The degree of transportation convenience of the pond X13 (0.067) | High degree—low degree | 1–5 |

**Table 1.** *Cont.*

| Target Layer | Criterion Layer | Indicator Layer | Definition | Value |
|---|---|---|---|---|
| | The urgency for construction land consolidation (0.213) | The urgency to improve rural infrastructure and public service facilities | The completeness of rural infrastructure X14 (0.306) | High degree of completeness—low degree of completeness | 1–5 |
| | | | The completeness of rural public service facilities X15 (0.363) | High degree of completeness—low degree of completeness | 1–5 |
| | | The urgency to achieve efficient use of rural construction land | The intensive utilization of rural construction land X16 (0.331) | High Use—low use | 1–5 |
| | The urgency for rural ecological protection and restoration (0.181) | The urgency for ecological environment restoration | The degree of water pollution X17 (0.085) | Low degree—high degree | 1–5 |
| | | | The degree of soil pollution X18 (0.071) | Low degree—high degree | 1–5 |
| | | | The degree of soil erosion X19 (0.100) | Low degree—high degree | 1–5 |
| | | | The degree of mine environment damage X20 (0.136) | Low damage—high damage | 1–5 |
| | | | The degree of vegetation degradation X21 (0.116) | Low degree—high degree | 1–5 |
| | | The urgency for human settlements improvement | The satisfaction degree of sanitary toilet renovation X22 (0.121) | High satisfaction—low satisfaction | 1–5 |
| | | | The satisfaction degree of domestic waste treatment X23 (0.160) | High satisfaction—low satisfaction | 1–5 |
| | | | The satisfaction degree of domestic sewage treatment X24 (0.101) | High satisfaction—low satisfaction | 1–5 |
| | | | The satisfaction degree of village appearance X25 (0.110) | High satisfaction—low satisfaction | 1–5 |
| | The urgency for rural historical and cultural protection (0.217) | The willingness to invest in rural historical and cultural protection | The richness of historical and cultural resources X26 (0.144) | Low degree—high degree | 1–5 |
| | | | The popularity of historical and cultural resources X27 (0.444) | Low popularity—high popularity | 1–5 |
| | | | The degree of destruction of historical and cultural relics X28 (0.197) | Low damage—high damage | 1–5 |
| | | | The willingness to build village historiographers X29 (0.215) | Low willingness— high willingness | 1–5 |
| | The urgency for industrial development (0.210) | The urgency for industrial development | The willingness to develop large-scale agriculture X30 (0.545) | Low willingness— high willingness | 1–5 |
| | | | The willingness to develop rural secondary and tertiary industries X31 (0.455) | Low willingness— high willingness | 1–5 |

### 2.3.2. Evaluation Index System from the Perspective of Social Investors

The core demand of social investors to invest in the comprehensive land consolidation projects is to obtain income. The main sources of income include two aspects. First, the balance index of cultivated land occupation and compensation generated by land consolidation and the balance index linked to the increase and decrease in urban and rural construction land are used for transactions so as to obtain income (hereinafter referred to as "index transaction income"). Second, use local resource endowments to develop industries and obtain income through industrial operations. Therefore, the more favorable the existing resource endowment in the project area is for industrial development, the more balance indicators that can be obtained through consolidation, the stronger the willingness

of social investors to invest in comprehensive land consolidation. The opposite is also true. Based on this, this paper constructs the comprehensive land consolidation projects decision-making evaluation index system from the perspective of social investors from the following five aspects.

The willingness to invest in agricultural land consolidation. In the process of agricultural land consolidation, the interests of social investors mainly include: the comprehensive quality of existing paddy fields, the potential of converting dry land to paddy fields, the comprehensive quality of other agricultural land, and the potential of new cultivated land from agricultural land consolidation and unused land development. In this paper, the willingness to invest in agricultural land consolidation is subdivided into the following four indicators: the willingness to invest in improving the comprehensive quality of existing paddy fields, the willingness to invest in transforming dry land into paddy fields, the willingness to invest in improving the comprehensive quality of other agricultural land, and the willingness to invest in new cultivated land. Among them, the willingness to invest in improving the comprehensive quality of existing paddy fields includes four indicators: the difficulty of improving paddy field irrigation facilities, the difficulty of improving paddy field drainage and waterlogging facilities, the difficulty of improving field road facilities, and the difficulty of reducing paddy field fragmentation, which are negative indicators. The willingness to invest in transforming dry land into paddy fields is represented by the difficulty of transforming dry land into paddy field, which is a negative index. The willingness to invest in improving the comprehensive quality of other agricultural land includes the willingness to invest in improving the comprehensive quality of garden land, economic forest land, and the pond. The willingness to invest in improving the comprehensive quality of garden land is measured by the difficulty of improving the irrigation facilities of the garden land and the degree of transportation convenience of the garden land. The willingness to invest in improving the comprehensive quality of economic forest land is measured by the difficulty of improving the irrigation facilities of economic forest land and the degree of transportation convenience of economic forest land. The willingness to invest in improving the comprehensive quality of the pond is measured by the difficulty of cleaning up, the difficulty of repairing leakages, the convenience of irrigation, and the degree of transportation convenience. Except for the difficulty of improving paddy field irrigation facilities, the difficulty of improving paddy field drainage and waterlogging facilities, the difficulty of cleaning up ponds, and the difficulty of repairing leakages of pond, which are negative indicators, all the other indicators are positive indicators. The willingness to invest in new cultivated land is measured by the proposed new cultivated land area of farmland consolidation and unused land development, which is a positive indicator.

The willingness to invest in construction land consolidation. In the process of construction land consolidation, the interests of social investors mainly include: the status of rural infrastructure, the status of rural public service facilities, and the potential of rural inefficient construction land reclamation. This paper subdivides the willingness to invest in construction land consolidation into the following two indicators: the willingness to invest in improving rural infrastructure and public service facilities and the willingness to invest in reclamation of rural inefficient construction land. Among them, the willingness to invest in improving rural infrastructure and public service facilities is represented by the difficulty of improving rural infrastructure and rural public service facilities, which are negative indicators. The willingness to invest in reclamation of rural inefficient construction land is represented by the proposed new cultivated land area of rural cultivated land reclamation, which is a positive indicator.

The willingness to invest in rural ecological protection and restoration. In the process of rural ecological protection and restoration, the interests of social investors mainly include: ecological environment and human settlements. In this paper, the willingness to invest in rural ecological protection and restoration is subdivided into two indicators: the willingness to invest in ecological environment restoration and the willingness to invest in human settlements improvement. Among them, the willingness to invest in ecological

environment restoration is represented by the difficulty of water pollution restoration, soil pollution restoration, soil erosion restoration, mine environment restoration, and vegetation degradation restoration, all of which are negative indicators. The willingness to invest in human settlements improvement is represented by the satisfaction degree of sanitary toilet renovation, domestic waste treatment, domestic sewage treatment, and village appearance, all of which are positive indicators.

The willingness to invest in rural historical and cultural protection. The richer and more famous the local historical and cultural resources, the more willing social investors are to invest to develop the rural leisure tourism industry. This paper subdivides the willingness to invest in rural historical and cultural protection into the following three indicators: the richness of historical and cultural resources, the popularity of historical and cultural resources, and the difficulty of restoration of historical and cultural relics. The former two are positive indicators, and the latter is a negative indicator.

The willingness to invest in industrial development. In terms of industrial development, the interests of social investors mainly include: the superiority of tourism resources in the project area and the industrial foundation of the project area. In this paper, the willingness to invest in industrial development is subdivided into the following two indicators: the willingness to invest in tourism development and the willingness to invest in industrial scale expansion and quality improvement. Among them, the willingness to invest in tourism development is represented by the superiority of tourism resources, which is a positive indicator. The willingness to invest in the industrial scale expansion and quality improvement is represented by the popularity of industrial operators, the popularity of characteristic industries, and the inclusion level of characteristic industries in the planning, all of which are positive indicators.

To evaluate the decision-making of the comprehensive land improvement project from the perspective of social investors is to judge the priority of the project by measuring the willingness of social investors to invest in comprehensive land consolidation. The higher the willingness of social investors to invest in projects, the higher the order of project approval. In order to accurately measure the willingness of social investors to invest, this paper adopts the Likert 5-level scale as a tool to measure the willingness. The specific method is shown in Table 2.

### 2.3.3. Evaluation Index System from the Comprehensive Perspective of Farmers and Social Investors

The decision-making evaluation of comprehensive land consolidation projects from the comprehensive perspectives of farmers and social investors is to combine the previous evaluation of the urgency of farmers' needs for comprehensive land consolidation and the willingness of social investors to invest to comprehensively determine the priority of comprehensive land consolidation, and then determine the priority order of pilot application projects. The higher the urgency of farmers' needs and the willingness of social investors to invest, the higher the priority of comprehensive land consolidation projects, and the higher the order of project approval. See Table 3 for details of the decision-making evaluation index system of comprehensive land consolidation projects from the perspectives of farmers and social investors.

Table 2. Decision-making evaluation index system of pilot projects of comprehensive land consolidation from social investors' perspective.

| Target Layer | Criterion Layer | Indicator Layer | Definition | Value |
|---|---|---|---|---|
| The willingness of social investors to invest in comprehensive land consolidation | The willingness to invest in agricultural land consolidation (0.335) | The willingness to invest in improving the comprehensive quality of existing paddy fields | The difficulty of improving paddy field irrigation facilities Y1 (0.068) | High difficulty—low difficulty | 1–5 |
| | | | The difficulty of improving paddy field drainage and waterlogging facilities Y2 (0.087) | High difficulty–low difficulty | 1–5 |
| | | | The difficulty of improving field road facilities Y3 (0.043) | High difficulty—low difficulty | 1–5 |
| | | | The difficulty of reducing paddy field fragmentation Y4 (0.049) | High difficulty—low difficulty | 1–5 |
| | | The willingness to invest in transforming dry land into paddy fields | The difficulty of transforming dry land into paddy field Y5 (0.216) | High difficulty—low difficulty | 1–5 |
| | | The willingness to invest in improving the comprehensive quality of other agricultural land | The difficulty of improving the irrigation facilities of the garden land Y6 (0.083) | High difficulty—low difficulty | 1–5 |
| | | | The degree of transportation convenience of the garden land Y7 (0.044) | Low degree—high degree | 1–5 |
| | | | The difficulty of improving the irrigation facilities of economic forest land s Y8 (0.095) | High difficulty—low difficulty | 1–5 |
| | | | The degree of transportation convenience of economic forest land Y9 (0.087) | Low degree—high degree | 1–5 |
| | | | The difficulty of cleaning up ponds Y10 (0.044) | High difficulty—low difficulty | 1–5 |
| | | | The difficulty of repairing leakages of the pond Y11 (0.043) | High difficulty—low difficulty | 1–5 |
| | | | The convenience of irrigation of the pond Y12 (0.037) | Low degree—high degree | 1–5 |
| | | | The degree of transportation convenience of the pond Y13 (0.042) | Low degree—high degree | 1–5 |
| | | The willingness to invest in new cultivated land | The proposed new cultivated land area of farmland consolidation and unused land development Y14 (0.062) | Small area—large area | 1–5 |

**Table 2.** *Cont.*

| Target Layer | Criterion Layer | | Indicator Layer | Definition | Value |
|---|---|---|---|---|---|
| | The willingness to invest in construction land consolidation (0.121) | The willingness to invest in improving rural infrastructure and public service facilities | The difficulty of improving rural infrastructure Y15 (0.325) | High difficulty—low difficulty | 1–5 |
| | | | The difficulty of improving rural public service facilities Y16 (0.354) | High difficulty—low difficulty | 1–5 |
| | | The willingness to invest in reclamation of rural inefficient construction land | The proposed new cultivated land area of rural cultivated land reclamation Y17 (0.321) | Small area—large area | 1–5 |
| | The willingness to invest in rural ecological protection and restoration (0.179) | The willingness to invest in ecological environment restoration | The difficulty of water pollution restoration Y18 (0.090) | High difficulty—low difficulty | 1–5 |
| | | | The difficulty of soil pollution restoration Y19 (0.110) | High difficulty—low difficulty | 1–5 |
| | | | The difficulty of soil erosion restoration Y20 (0.093) | High difficulty—low difficulty | 1–5 |
| | | | The difficulty of mine environment restoration Y21 (0.116) | High difficulty—low difficulty | 1–5 |
| | | | The difficulty of mine environment restoration Y22 (0.107) | High difficulty—low difficulty | 1–5 |
| | | The willingness to invest in human settlements improvement | The satisfaction degree of sanitary toilet renovation Y23 (0.087) | Low satisfaction—high satisfaction | 1–5 |
| | | | The satisfaction degree of domestic waste treatment Y24 (0.107) | Low satisfaction—high satisfaction | 1–5 |
| | | | The satisfaction degree of domestic sewage treatment Y25 (0.099) | Low satisfaction—high satisfaction | 1–5 |
| | | | The satisfaction degree of village appearance Y26 (0.191) | Low satisfaction—high satisfaction | 1–5 |
| | The willingness to invest in rural historical and cultural protection (0.208) | The willingness to invest in rural historical and cultural protection | The richness of historical and cultural resources Y27 (0.185) | Low degree—high degree | 1–5 |
| | | | The popularity of historical and cultural resources Y28 (0.570) | Low popularity—high popularity | 1–5 |

**Table 2.** *Cont.*

| Target Layer | Criterion Layer | | Indicator Layer | Definition | Value |
|---|---|---|---|---|---|
| | | | The difficulty of restoration of historical and cultural relics Y29 (0.245) | High difficulty—low difficulty | 1–5 |
| | The willingness to invest in industrial development (0.157) | The willingness to invest in tourism development | The superiority of tourism resources Y30 (0.386) | low superiority—high superiority | 1–5 |
| | | The willingness to invest in industrial scale expansion and quality improvement | The popularity of industrial operators Y31 (0.192) | Low popularity—high popularity | 1–5 |
| | | | The popularity of characteristic industries Y32 (0.271) | Low popularity—high popularity | 1–5 |
| | | | The inclusion level of characteristic industries in the planning Y33 (0.151) | Low level—high level | 1–5 |

**Table 3.** Decision-making evaluation index system of pilot projects of comprehensive land consolidation from the comprehensive perspective of farmers and social investors.

| Target Layer | Criterion Layer | Indicator Layer | Definition |
|---|---|---|---|
| The priority of comprehensive land consolidation | The urgency of farmers' needs for comprehensive land consolidation (0.307) | The urgency for agricultural land consolidation (0.179) | X1–X13 |
| | | The urgency for construction land consolidation (0.213) | X14–X16 |
| | | The urgency for rural ecological protection and restoration (0.181) | X17–X25 |
| | | The urgency for rural historical and cultural protection (0.217) | X26–X29 |
| | | The urgency for industrial development (0.210) | X30–X31 |
| | The willingness of social investors to invest in comprehensive land consolidation (0.693) | The willingness to invest in agricultural land consolidation (0.335) | Y1–Y14 |
| | | The willingness to invest in construction land consolidation (0.121) | Y15–Y17 |
| | | The willingness to invest in rural ecological protection and restoration (0.179) | Y18–Y26 |
| | | The willingness to invest in rural historical and cultural protection (0.208) | Y27–Y29 |
| | | The willingness to invest in industrial development (0.157) | Y30–Y33 |

## 3. Results

### 3.1. Decision-Making Evaluation Results of Comprehensive Land Consolidation Projects from Two Separate Perspectives

After sorting out the valid sample data and processing it through the simple arithmetic average method, the entropy weight TOPSIS method was used to carry out a quantitative analysis on the urgency of farmers' needs and the willingness of social investors to invest in the seven declared projects and to obtain decision-making evaluation results of comprehensive land consolidation projects from different perspectives. The results are shown in Table 4.

As can be seen from Table 4, from the perspective of the urgency of farmers' needs for comprehensive land consolidation, projects A and C should be established first, followed by projects G, F, E, D, and B. However, from the perspective of investment willingness of social investors, the order of project approval is C, E, D, F, A, G, and B. It can be seen that there is a big difference in the evaluation results of project approval from the perspective of farmers and social investors, mainly due to the different interests and concerns of farmers and social investors. Farmers are the masters of the village and the ultimate beneficiaries of comprehensive land consolidation. Compared with the index benefits brought by comprehensive land consolidation, they pay more attention to the consolidation content closely related to their own production and life, such as agricultural land consolidation to improve the quality of cultivated land, rural ecological protection and restoration to improve the quality of human settlements, and development of large-scale agriculture and industrial integration. As an investor, the core appeal of social investors participating in comprehensive land consolidation is to obtain index transaction income and industrial operation income. Therefore, compared with the interests of farmers, social investors pay more attention to the tradable surplus indicators provided by comprehensive land consolidation and the advantageous resources supporting the development of rural industries, such as beautiful ecological environment, rich historical and cultural resources, tourism resources and industrial base, etc.

**Table 4.** Decision-making evaluation results of the comprehensive land consolidation projects from two separate perspectives.

| Declaration Project | Farmers' Perspective | | Social Investors' Perspective | |
|---|---|---|---|---|
| | Urgency of Need | Sort | Willingness to Invest | Sort |
| Zhaoliqiao Town Project in Chibi City (A) | 0.540 | 1 | 0.373 | 5 |
| Henggouqiao Town Project in Xian'an District (B) | 0.307 | 7 | 0.278 | 7 |
| Xiangyanghu Town Project in Xian'an District (C) | 0.540 | 1 | 0.715 | 1 |
| Dupu Town Project in Jiayu County (D) | 0.446 | 6 | 0.418 | 3 |
| Daping Township Project in Tongcheng County (E) | 0.451 | 5 | 0.455 | 2 |
| Tiancheng Town Project in Chongyang County (F) | 0.469 | 4 | 0.414 | 4 |
| Honggang Town Project in Tongshan County (G) | 0.489 | 3 | 0.314 | 6 |

From the above analysis, it can be seen there will be great differences in the evaluation results when the decision-making of comprehensive land consolidation pilot projects is carried out solely from the perspectives of farmers and social investors. Therefore, the decision-making evaluation of comprehensive land consolidation project should comprehensively consider the interests of farmers and social investors in order to make the decision-making evaluation results more scientific and reasonable.

*3.2. Decision-Making Evaluation Results of Comprehensive Land Consolidation Projects from the Comprehensive Perspective of Farmers and Social Investors*

To combine the evaluation results of the urgency of farmers' needs and the willingness of social investors to invest in the seven declared projects, the entropy weight TOPSIS method was also used to obtain the decision-making evaluation results of comprehensive land consolidation projects from the perspective of farmers and social investors. The results are shown in Table 5.

**Table 5.** Decision-making evaluation results of the comprehensive land consolidation projects from the comprehensive perspective of farmers and social investors.

| Declaration Project | Based on the Perspective of Farmers and Social Investors | | Actual Project Results |
|---|---|---|---|
| | Priority | Sort | |
| Zhaoliqiao Town Project in Chibi City (A) | 0.424 | 5 | Not approved |
| Henggouqiao Town Project in Xian'an District (B) | 0.287 | 7 | Municipal pilot project |
| Xiangyanghu Town Project in Xian'an District (C) | 0.661 | 1 | Provincial pilot projects |
| Dupu Town Project in Jiayu County (D) | 0.427 | 4 | Not approved |
| Daping Township Project in Tongcheng County (E) | 0.454 | 2 | Provincial pilot projects |
| Tiancheng Town Project in Chongyang County (F) | 0.431 | 3 | Provincial pilot projects |
| Honggang Town Project in Tongshan County (G) | 0.368 | 6 | Not approved |

As can be seen from Table 5, regarding the priority of declared projects for comprehensive land consolidation from the perspective of farmers and social investors, the C project (0.661) should be given priority, followed by the E project (0.454), followed by the F project (0.431), D item (0.427), A item (0.424), G item (0.368), and B item (0.287). The above results comprehensively consider the urgency of farmers' needs and the willingness of social investors to invest in the comprehensive land consolidation, and the evaluation results are basically consistent with the actual project establishment. Among them, projects C, E and F, which are the top three in the priority ranking of the declared projects for comprehensive land consolidation, were identified as the provincial pilot project in the comprehensive land consolidation project decision-making evaluation organized by the Office of the Leading Group for Comprehensive Land Consolidation in Hubei Province in 2020. However, the B project with the lowest priority was identified as a municipal pilot project, while the D project with the fourth priority and the A project with the fifth priority

were not approved. It can be seen that the current comprehensive land consolidation project decision-making evaluation in Hubei Province basically considers the interests of farmers and social investors, which are two important subjects, and generally seems to be reasonable. However, there is still room for further improvement. No matter whether from the perspective of farmers or social investors, project B was ranked last in the order of project approval, but it was listed as a municipal pilot project, which shows that there is a certain deviation in comprehensive land consolidation project decision-making evaluation in Hubei Province. In the future, we should comprehensively consider the urgency of farmers' needs and the willingness of social investors to invest and finally determine the priority of comprehensive land consolidation projects. This will not only safeguard the rights of farmers but also leverage the participation of social capital and, finally, ensure the smooth implementation of comprehensive land consolidation projects.

## 4. Discussion

### 4.1. Project Priority Analysis

On the whole, the priority order evaluation results of comprehensive land consolidation project from the comprehensive perspective of farmers and social investors are basically consistent with the actual project establishment. Whether it is the urgency of farmers' needs or the willingness of social investors to invest, Project C has the highest score, and Project C has also been confirmed as a provincial pilot project in reality. Through field research, it was found that C project area state-owned farms accounted for 44.18% of the total area. State-owned farms have the advantage of mechanization and organization, which is conducive to the development of agricultural modernization and industrial management. The project area has already invested in two beautiful countryside projects, two water conservancy projects, and five land consolidation projects, which have effectively improved the human settlements, ecological environment, and cultivated land quality. The project area is rich in cultural resources, and the former site of cultural celebrities in Xiangyanghu was listed in the seventh batch of national key cultural relics protection unit in 2013. The farm advantages, environmental advantages, industrial advantages, and cultural advantages of Project C made it listed as a provincial pilot project.

However, there is a certain deviation between the priority of the B project and the actual project approval result. Project B is located in a provincial modern agricultural industrial park and was listed as a provincial pilot project in the comprehensive land consolidation of Hubei Province in 2020. Still, no matter whether from the perspective of farmers or social investors, the order of project B is ranked last. Through the investigation, it was found that the population outflow in the B project area is very serious, and there are many "empty nests" of young people going out with the elderly and children staying behind, which makes less the urgency of farmers' needs for comprehensive land consolidation in the project area. The industrial parks in the project area are in pursuit of economic benefits, ignoring infrastructure construction, seriously restricting the further development of the industrial base in the project area, which is also the reason for the low investment willingness of social investors.

In general, the inconsistency between the decision-making priorities and the actual project approval results shows that there is a certain deviation in the current comprehensive land consolidation project decision-making evaluation in Hubei Province. Generally speaking, the government is willing to invest in villages with better existing resource endowments, while farmers and social investors have different concerns. For farmers, the worse the existing agricultural land, construction land, and ecological environment in the countryside, the higher the urgency of farmers' needs. Social investors pay more attention to the tradable surplus indicators provided by comprehensive land consolidation and the future development potential of the countryside.

### 4.2. Research on Decision-Making of Comprehensive Land Consolidation Project in Other Countries

Land consolidation requires difficult and conflicting decisions such as where to revitalize the declining countryside [11]. In many European countries, especially those receiving European Union (EU) support for land consolidation projects, it is important to carefully allocate funds to the most suitable areas [30]. Traditionally, these decisions have been made by groups, some linked to the area being consolidated and others from the government, all of whom attempt to create the best possible decision [2]. During a comprehensive literature analysis and interviews with land consolidation experts, it was noted that certain countries, such as Finland, use country-wide maps to identify potential areas for land consolidation. Some countries undertake various marketing activities, information campaigns, and other methods to raise public awareness. One of the recent examples is the Dutch Kadaster, which celebrated 100 years of practice in implementing land consolidation projects in 2016 with the release of Move a Lot, a smart device game that allows players to "play" to re-adjust land consolidation project areas. This approach can mobilize the enthusiasm of farmers, maximize the protection of farmers' rights, and earn the support of active local leaders—"social activists" [31]. Recent studies have highlighted the need to identify the most suitable and prioritized land consolidation areas at different levels of governance [32–36]. However, the scale and the criteria vary from country to country and are influenced by the national as well as regional policies and strategies.

In a word, the establishment of comprehensive land consolidation projects is a common problem all over the world. To learn how to invest the limited renovation funds into the most suitable and leading regions, countries need to explore the most suitable road for themselves.

### 4.3. Deficiencies and Suggestions for Improvement

The construction of the comprehensive land consolidation project decision-making evaluation index system from the perspectives of farmers and social investors is a supplement to the current independent policy decision-making. However, there are still the following deficiencies: (1) This paper only considers the interests of farmers and social investors to establish a comprehensive land consolidation project decision-making evaluation index system. However, the comprehensive land consolidation also involves multiple stakeholders such as local governments and village collectives. In the future, a comprehensive land consolidation project decision-making evaluation index system coordinated by multiple stakeholders should be established. (2) In order to avoid possible problems in the decision-making stage of project approval, a dynamic adjustment mechanism for pilot projects of comprehensive land consolidation should be established in the future, removing from the pilot list those that have been established but are unable to be implemented or have poor results. Moreover, the municipal pilot projects with good implementation effects will be adjusted to provincial pilot projects to obtain the support of provincial financial funds.

### 5. Conclusions

From the perspectives of farmers and social investors, this paper builds a comprehensive land consolidation project decision-making evaluation index system and makes an empirical analysis by using the survey data of seven pilot projects of comprehensive land consolidation in Xianning, Hubei Province, in 2020. Finally, the following research conclusions were obtained:

There will be great differences in the evaluation results when the decision-making of comprehensive land consolidation pilot projects is carried out solely from the perspectives of farmers and social investors. The reason is that farmers and social investors have different interests and concerns. Farmers pay more attention to whether comprehensive land consolidation can improve their production and living conditions, promote industrial development, and increase income. Social investors pay more attention to whether com-

prehensive land consolidation can produce indicators transaction income and industrial operating income.

It is reasonable and feasible to establish a comprehensive land consolidation project decision-making evaluation system when considering the interests of farmers and social investors. Social investors are the main investors in the comprehensive land consolidation, and farmers are the ultimate beneficiaries of the comprehensive land consolidation. Both are the core stakeholders of the comprehensive land consolidation. The decision-making evaluation of the comprehensive land consolidation project should comprehensively consider the interests and demands of farmers and social investors to make the evaluation results more scientific and reasonable.

The limited government financial funds should be invested in the consolidation content that the farmers need very much and social investors are willing to invest in. There is a large demand for funds for the comprehensive land consolidation project, which requires not only government financial capital investment but also a large amount of social capital investment. Thus, in the early stage of the pilot work of comprehensive land consolidation, limited government financial funds should be invested in places where farmers are in great need of improvement and in which social investors are willing to invest. In this way, it can not only attract social capital to participate in the comprehensive land consolidation so as to solve the current imbalance between the supply and demand of funds for the comprehensive land consolidation but also truly enhance the sense of gain of the farmers in the comprehensive land consolidation project area and promote common prosperity.

**Author Contributions:** Conceptualization, W.X. and G.Y.; methodology, W.X.; software, W.X.; validation, W.X. and G.Y.; formal analysis, W.X.; investigation, W.X.; resources, W.X.; data curation, W.X.; writing—original draft preparation, W.X.; writing—review and editing, W.X.; visualization, W.X. and G.Y.; supervision, G.Y.; project administration, G.Y.; funding acquisition, G.Y. All authors have read and agreed to the published version of the manuscript.

**Funding:** This research was funded by the National Natural Science Foundation of China, grant number 71904150, and the National Social Science Foundation of China, grant number 12BGL078.

**Institutional Review Board Statement:** Not applicable.

**Informed Consent Statement:** Informed consent was obtained from all subjects involved in the study.

**Data Availability Statement:** Not applicable.

**Conflicts of Interest:** The authors declare no conflict of interest.

## Appendix A

**Table A1.** Overview of each application project in the study area.

| Project | A | B | C | D | E | F | G |
|---|---|---|---|---|---|---|---|
| Geographic location | The south of Zhaoliqiao Town in Chibi City | The south of Henggouqiao Town in Xian'an District | The east of Xiangyanghu Town in Xian'an District | The south of Dupu Town in Jiayu County | The south of Daping Township in Tongcheng County | The northern frontier of Tiancheng Town in Chongyang County | The southwest of Honggang Town in Tongshan County |
| Geographic type | Low mountains and hills | Low mountains and hills | Gentle slope plain | Plains and hills | Downland | Basin | Low mountains and hills |
| Population of Project Area (person) | 6870 | 6442 | 4884 | 14,793 | 8986 | 7983 | 4954 |
| Project Area (ha) | 3958.57 | 2141.37 | 2260.11 | 8323.70 | 1195.70 | 5261.22 | 5849.22 |
| Cultivation area (ha) | 359.09 | 641.87 | 501.13 | 1368.58 | 597.79 | 308.89 | 202.65 |
| Per capita income (yuan) | 11,358 | 16,655 | 24,414 | 18,873 | 16,361 | 19,794 | 7500 |

**Table A2.** Survey of individual characteristics of interviewed farmers.

x

| Content | Classification | Sample Size (Copies) | Proportion (%) |
|---|---|---|---|
| Gender | Man | 265 | 86.60 |
| | Woman | 41 | 13.40 |
| Age (years) | <40 | 7 | 2.29 |
| | [40, 50) | 34 | 11.11 |
| | [50, 60) | 104 | 33.99 |
| | [60, 70) | 99 | 32.35 |
| | ≥70 | 62 | 20.26 |
| Level of education | Illiteracy | 46 | 15.03 |
| | Primary school | 124 | 40.52 |
| | Junior high school | 98 | 32.03 |
| | High school or technical secondary school | 34 | 11.11 |
| | College degree and above | 4 | 1.31 |
| Whether the village is cadre | Yes | 6 | 2.00 |
| | No | 300 | 98.00 |
| Types of employment | Agricultural production | 166 | 54.24 |
| | Local business | 16 | 5.23 |
| | Local workers | 76 | 24.84 |
| | Nonlocal business | 0 | 0 |
| | Nonlocal workers | 48 | 15.69 |

**Table A3.** Overview of social investors interviewed.

| Survey Content | Sorting Criterion | Number of Samples (Copies) | Sample Proportion (%) |
|---|---|---|---|
| New types of business entities | Professional investors | 2 | 10.00 |
| | Family farm | 1 | 5.00 |
| | Farmers' professional cooperative | 7 | 35.00 |
| | Corporate champion | 10 | 50.00 |
| Type of industry | Primary industry | 10 | 50.00 |
| | Secondary industry | 7 | 35.00 |
| | Tertiary industry | 1 | 5.00 |
| | Integration of primary and secondary industries | 1 | 5.00 |
| | Integration of primary and tertiary industries | 1 | 5.00 |
| | Integration of secondary and tertiary industries | 0 | 0 |
| Annual output value (ten thousand yuan) | ≤50 | 7 | 35.00 |
| | [50, 100) | 4 | 20.00 |
| | [100, 500) | 6 | 30.00 |
| | [500, 1000) | 1 | 5.00 |
| | ≥1000 | 2 | 10.00 |

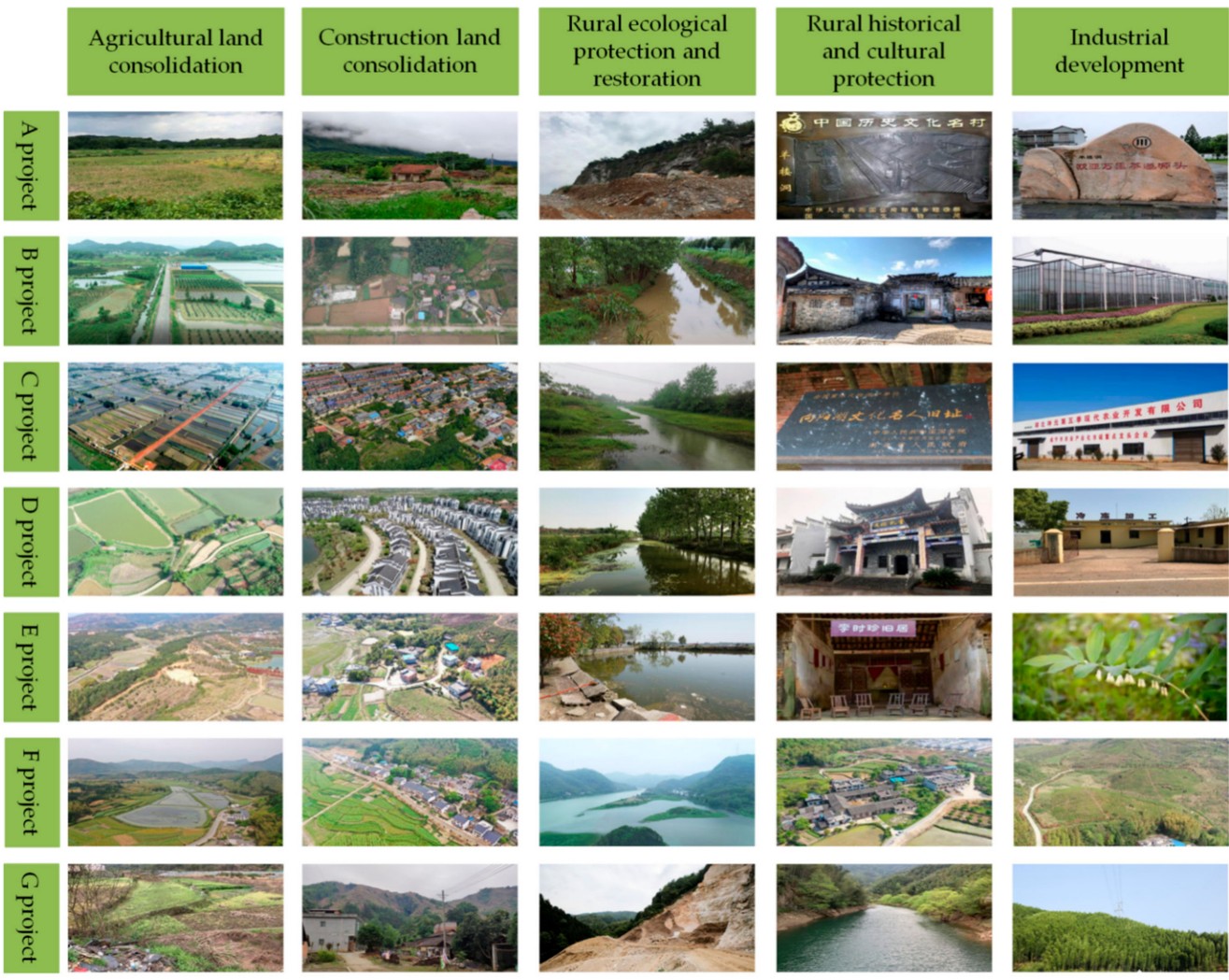

**Figure A1.** Basic information of the project area.

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
