# Peer review of "Decision-Making Evaluation of the Pilot Project of Comprehensive Land Consolidation from the Perspective of Farmers and Social Investors: A Case Study of the Project Applied in Xianning City, Hubei Province, in 2020"

_land, doi:10.3390/land11091534_

Round 1

Reviewer 1 Report

It is a very good research that can support decision making in LUC project. 

I have a very minor comment: 

Sections 3.1 and 3.2 have similar titles. It will be good to fid a new title for 3.2 and put the one of 3.2. to 3.1. They are only 2 perspectives in this research. I think it can be misleading to say 'different perspectives' while they are only two. 

Author Response

Dear Reviewers,

Thank you for your letter and comments concerning our manuscript entitled " Decision-making evaluation of the pilot project of comprehensive land consolidation from the perspective of farmers and social investors: A case study of the project applied in Xianning City, Hubei Province in 2020" (land- 1905625).

The comments were valuable for improving our manuscript and guiding our research. We have studied the comments carefully and have made revisions accordingly. We hope that the revisions meet with your approval and please see the attachment.

Reviewer 2 Report

Dear Authors,

The title of the study “Decision-making evaluation of the pilot project of comprehensive land consolidation from the perspective of farmers and social investors— A case study of the project applied in Xianning City, Hubei Province in 2020” corresponds to its content.

Keywords: comprehensive land consolidation; pilot project; decision-making evaluation; farmers; social investors are correct.

The total value of work is a valuable contribution. References take 28 publications are cited in the entire article. Literature research well started, but not enough publications. It is proposed to add the following articles that contain new research in this area, for example:

·         StrÄ™k, Å».; Noga, K. 2019. Method of delimiting the spatial structure of villages for the purposes of land consolidation and exchange. Remote Sens. 2019, 11, 1268. https://doi.org/10.3390/rs11111268

·         Balawejder, M., Matkowska, K., & Rymarczyk, E. 2021. Effects of land consolidation in Southern Poland. Acta Scientiarum Polonorum Administratio Locorum, 20(4), 269–282. https://doi.org/10.31648/aspal.6573

Similarly, the discussion or conclusion should refer to research conducted in this field in other countries and cited in this publication. Please complete this and the article will be a valuable scientific contribution.

Figure 1. Survey area. - not readable (blurry). Please improve the quality of figure 1. For figure 1.3, please correct the names in Chinese to the English names.

It should be noted that the whole of the study is cognitive and contains important scientific elements. The article was written at a good academic level. In relation to the above, I express the opinion that the work submitted for review should be published in its entirety after taking into account the comments of the reviewer but not require a review again.

Author Response

(The authors gave the same response as above.)
